

Shear rate effect on the residual strength characteristics of saturated loess
Baoqin Lian[a,b], Jianbing Peng[a*], Qiangbing Huang[a]
[a]College of Geological Engineering and Surveying, Chang'an University, Key
Laboratory of Western China Mineral Resources and Geological Engineering, Xi'an
710054, China
[b]Department of Geology & Geophysics, Texas A&M University, College Station, TX
77843-3115, United States
*Corresponding author: Jianbing Peng (dicexy1@gmail.com)




**Abstract**

Residual shear strength of soils is an important soil parameter for assessing the
stability of landslides. To investigate the effect of the shear rate on the residual shear
strength of loessic soils, a series of ring shear tests were carried out on loess from
three landslides at two shear rates (0.1 mm/min and 1 mm/min). Naturally drained
ring shear tests results showed that the shear displacement to achieve the residual
stage for specimens with higher shear rate was greater than that of the lower rate; both
the peak and residual friction coefficient became smaller with increase of shear rate
for each sample; at two shear rates, the residual friction coefficients for all specimens
under the lower normal stress were greater than that under the higher normal stress.
The tests results revealed that the difference in the residual friction angle $\phi_r$ at the two
shear rates, $\phi_r$ (1)- $\phi_r$ (0.1), under each normal stress level were either positive or
negative values. However, the difference $\phi_r$ (1) - $\phi_r$ (0.1) under all normal stresses
was negative, which indicates that the residual shear parameters reduced with the
increasing of the shear rate in loess area. Such negative shear rate effect on loess
could be attributed to a greater ability of clay particles in specimen to restore broken
bonds at low shear rates.

**Keywords:** Loess; Residual shear strength; Ring shear test; Shear rate; Residual shear
parameter




## 1.  Introduction

Residual shear strength of soil is of great significance for evaluating the stability
for the slip surface of first-time landslides as well as reactivated landslides (Bishop et
al., 1971; Mesri and Shahien, 2003). The residual strength of soils is defined as the
minimum constant value of strength along the slip plane, in which the soil particles
are reoriented and subjected to sufficiently large displacements in relatively low shear
rate (Skempton, 1985).
Numerical studies have been done to assess the residual strength through the
laboratory tests using ring shear tests and reversal direct shear tests (Moeyersons et al.,
2008; Summa et al., 2010; Vithana et al., 2012; Chen and Liu, 2013; Summa et al.,
2018). It is a generally accepted fact that the measurement of the residual strength is
most preferred done with a ring shear test since it allows the soil specimen be sheared
at unlimited displacement which can simulate the field conditions more accurately
(Lupini et al., 1981; Sassa et al., 2004; Tiwari and Marui, 2005; Bhat, 2013). Until
now, great efforts have been paid to the study of the shear rate effect on the minimum
value of clay or sand strength at residual states (Morgenstern and Hungr, 1984; Lemos,
1985; Tika, 1999; Tika and Hutchinson, 1999; Suzuki et al., 2007; Grelle and
Guadagno, 2010; Bhat, 2013). As a result, the residual strength of clay or sand under
the effect of shear rate has been made relatively clear. However, compared with the
results of tests on clay or sand, understanding of the shear characteristics of silty soil,
such as loess, is not yet complete. As pointed out by Ding (2016), some drained ring
shear tests have concluded that the increase in shear rate causes the residual strength



of loess to increase. On the contrary, Kimura et al. (2014) reported that the residual
strength of Malan loess decreases with the increase of shear rate. Furthermore, Wang
et al. (2015) found that the effect of shear rate on residual strength of loess is closely
associated with the normal stress levels, and the change in residual strength of loess
samples under high normal stress levels is small in ring shear tests.

Therefore, some inconsistent or even opposite results have been reported in the

ring shear tests on loess above, which implied that there is still a lack of experimental
data on this topic. From the above investigations, it can be concluded that the effect of
the shear rate on the residual strength of the loess is not fully understood and needs
further scrutiny. Meanwhile, almost all of these investigations (Kimura et al., 2014;
Wang et al., 2015; Ding, 2016) focused on the residual shear characteristics of loess
obtained from the same location, while studies of loess collected from different
locations have only been rarely performed. Moreover, it should be noted that the
residual strength parameters (friction angle) obtained from using different shear rates
may be adopted to provide a guide for designing some precision engineering which
require high accuracy of the design parameters, thus, the effect of the shear rate on the
residual strength of soils should be fully investigated to determine the parameters with
high reliability. In addition, residual strength parameters of soil play a key role in
assessing the stability analysis of landslides. Therefore, accurate determination of the
residual strength parameters and their dependence on the shear rate may affect the
stability evaluation of landslides. Thus, it is necessary to study the change of residual
strength of loess with shear rate in order to have a good understanding of the suitable





approach for the residual strength parameters measurement.

In this backdrop, to clarify the residual shear characteristics of loess under the

effect of the shear rate, a series of naturally drained ring shear tests were conducted on
loess obtained from three landslides at two shear rates (0.1 mm/min and 1 mm/min).
The residual shear characteristics of loess at the residual state was examined.
Considering that shear strength of loess reduces with moisture content (Dijkstra et al.,
1994; Zhang et al., 2009; Picarelli, 2010), ring shear tests were conducted on
saturated loess samples corresponding to the worst condition in field engineering.
Furthermore, this study investigated the change in the residual strength parameters of
loess at different shear rates and their relationships with the normal stress in naturally
drained ring shear tests as well.

**2.   Geological setting of landslide sites**

Soil samples from three landslides in the northwest of China were selected in this

study. Soil samples used for the ring shear tests and index measuring tests
predominantly consist of loess deposits and were collected in a disturbed condition.
For convenience, the names of landslide sites were abbreviated into Djg, Ydg, and
Dbz. Fig. 1 shows the study sites and some views of the landslides.
**Dingjiagou landslide (Djg)**

The Djg landslide, located at the mouth of Dingjia Gully in Yan'an of China, is

geologically composed of upper loess and lower sand shale in the Yan-chang
formation. The dustpan-shaped landslide is inclined to the east, with its inclination
75.85°. The landslide is 350 m in width, 180 m in length, 70 m in elevation. The



average thickness of slip mass is around 20 m, and the volume of landslide totaled
approximately 105 x $10^4$ $m^3$. The slip mass is mainly constituted by loess, whereas the
sliding bed consists of sand shale in Yan-chang formation. The thickness of the
sliding zone varied from 30 to 50 cm. The front lateral region of the main slide
section of the Djg landslide, where the sampling was performed, was found to be silty
clay.
**Yandonggou landslide (Ydg)**
The Ydg landslide, located in the Qiaogou town of Yan'an in Shaan xi province of
China. The top and the toe altitude of the landslide are about 1165 m and 1110 m
above the sea level, with the height difference between the toe and the top of landslide
about 55 m. The slides have well-developed boundaries with the main sliding
direction of 240° and slope angle of 30°. From the landslides profile, the sliding
masses from top to bottom were classified by late Pleistocene ($Q_3$) loess, Lishi ($Q_2$)
loess and clay soil, respectively. Multiple landslides had occurred in this site, and the
soil samples used in this study were collected from $Q_2$ loess stratum within the slide
ranged from 4.5 m to 18 m in height.
**Dabuzi landslide (Dbz)**
The Dbz landslide located in the middle part of Shaanxi province (about E
108°51'36" and N 34°28'48"), China, which is a semi-arid zone dominated by loessic
geology. In this region, the investigated site is classified as a typical loess tableland
with quaternary stratum. The sedimentary losses in this area are grey yellow, and the
exposure stratum in this area has been divided into two stratigraphic units, namely,
the upper Malan ($Q_3$) loess and the lower Lishi ($Q_2$) loess, of which the $Q_3$ loess is



younger. The $Q_3$ loess is closest to the surface and is up to approximately 12 m thick,
while the thickness of $Q_2$ loess may reach an upper limit of about 50 m (Leng et al.,
2018). The loess in this area have well-developed vertical joints (Sun et al., 2009).
The travel distance and the maximum width of the slip mass are roughly estimated to
be 122 m and 133 m, respectively. The armchair-shaped landslide shows an apparent
sliding plane, with an area of approximately 15,660 $m^2$ and about 66.25 m maximum
difference in elevation. The main direction of this landslide is approximately 355°.
The exposed side scarp of the landslide, where the sampling was done, was found to
be entirely in the $Q_2$ loess stratum.

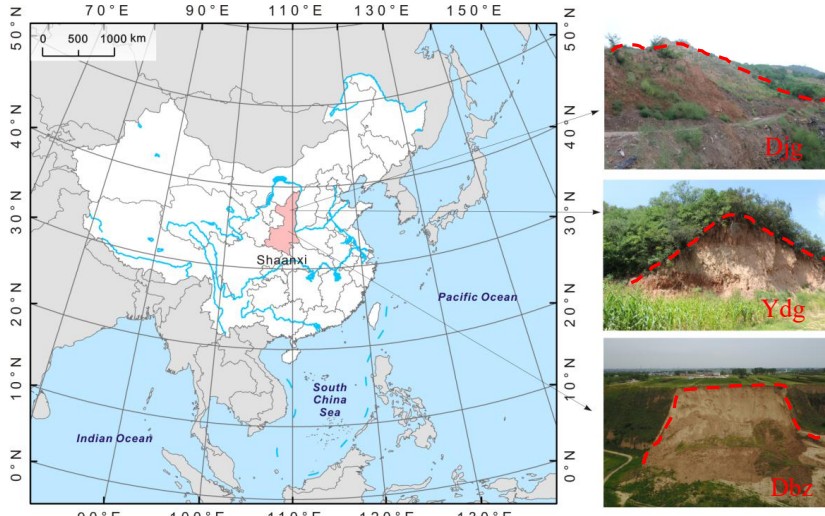


Figure 1. Location of study sites and some views of landslides
Notes: Red dashed lines in the Figure 1 represent landslide boundary.
**3.  Experimental scheme**
**3.1.  Testing sample**

The fact that the residual shear strength is independent of the stress history has

been reported by many researchers (Bishop et al., 1971; Stark Timothy et al., 2005;



Vithana et al., 2012). Thus, disturbed loess samples from each landslide weighing
about 25 kg were collected to investigate the residual shear strength.

The soil samples were air-dried, and then crushed with a mortar and pestle. It was

found that small lumps may exist in air-dried samples, which may be too big for the
cell, so lumps were crushed in order to make sample uniform. This should be done
with care so as not to destroy silty-dominated loess. After that, soil samples were
processed through 0.5 mm sieve. Distilled water was then added to the soil samples
until saturated water content were obtained. The physical parameters such as natural
moisture content (*in-situ* moisture content), specific gravity, bulk density, plastic limit,
and liquid limit were determined in accordance with the Chinese National Standards
(CNS) GB/T 50123-1999 (standards for soil test methods) (SAC, 1999), but clay size
was defined to be less than 2 *u*m followed ASTM, D 422 (ASTM, 2007). Each soil
sample was separated into clay (sub 0.002 mm), silt (0.002-0.075 mm), and sand
(0.075-0.5 mm) fractions. The physical indexes of the soil are listed in Table 1.

The grain size distribution of soil was measured using a laser particle size

analyzer Bettersize 2000 (Dandong Bettersize Instruments Corporation, Dandong,
China). The sieved soil samples were used to determine particle size distribution. In
this study, soil samples were treated with sodium hexaphosphate, serving as a
dispersant, to disaggregate the bond between the particles. The results show that the
clay fraction in Djg landslide soil (24%) is more than two times than that from Ydg
(9%) and Dbz (9.1%). Furthermore, the particle size analysis illustrated that the
percentage of silt-sized soil in three landslides ranged from 75.66% to 87.4%. In




addition, Ydg landslide soil consists of the greatest percentage of the sand fraction
which reaches up to 10.55%.
**Table 1** Physical parameters of slip-zone loess

| sites | $\rho_d$ | $W$ | $\rho$ | $G_S$ | $W_L$ | $W_p$ | Grain size fractions (%) | | | |
|---|---|---|---|---|---|---|---|---|---|---|
| | | | | | | | <0.002mm | 0.002-0.005mm | 0.005-0.075 | 0.075-0.5mm |
| Djg | 1.74 | 19.5 | 2.08 | 2.65 | 36 | 20 | 24 | 11.48 | 64.18 | 0.34 |
| Ydg | 1.47 | 18 | 1.74 | 2.71 | 33 | 19 | 9 | 5.28 | 75.17 | 10.55 |
| Dbz | 1.48 | 16 | 1.72 | 2.70 | 32 | 21 | 9.1 | 6.4 | 81 | 3.5 |

Notes: $\rho_d$= dry density (g/cm$^3$); w=moisture water content (%); $\rho$= bulk density
(g/cm$^3$); $G_S$ = specific gravity; W$_L$=liquid limit; $W_p$= plastic limit
**3.2.  Testing apparatus**

An advanced ring shearing apparatus (SRS-150) manufactured by GCTS (Arizona,

USA) was adopted in ring shear tests and the photos of apparatus were shown in Fig.
2, which consists mainly of a shear box with an outer diameter of 150 mm, an inter
diameter of 100 mm and the maximal sample height of 250 mm. The shearing box
consists of the upper shear box and the lower shear box. In the shearing process, the
upper shear box keeps still while the lower one rotates. The apparatus which provides
effective specimen area of 98 cm$^2$, is capable of shearing the specimen for large
displacements. The annular specimen is confined by inside and outside metal rings.
Moreover, the specimen is confined by bottom annular porous plates and top annular
porous plates in which have sharp-edged radial metal fins which protrude vertically
into the top and bottom of the specimen at the shearing process. Two annual porous
plates were used to provide drainage condition in the test following previous research
(Stark and Vettel, 1992). The normal stress, shear strength and shear displacement can



be monitored by computer in shearing process. The measurement features of the ring
shear apparatus employed in this study are described as follows: shear rate range from
0.001 degrees to 360 degrees per minute, 10 kN axial load capacity, 300 N. m
continuous torque capacity, maximum normal stress of 1000 kN/m$^2$.

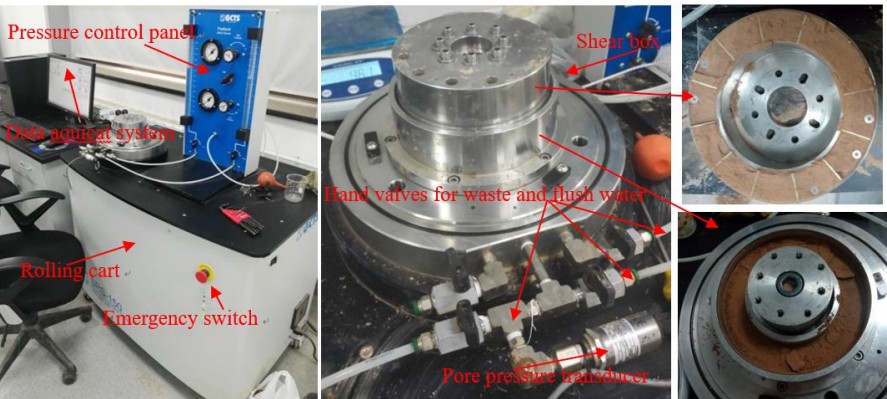


Figure 2. Ring shear apparatus (SRS-150)

**3.3.  Testing procedure**
In present study, reconstituted samples of the sub 0.5 mm soil fractions were used
in the testing as it was reported that the residual strength of the soil was unaffected by
its initial structure (Bishop et al., 1971; Vithana et al., 2012). Specimens were first
prepared by adding distilled water to the air-dried soil until the saturated moisture
contents were obtained. Then, specimens were kept in a sealed container for at least
one week to fully hydrate. Afterwards, specimens are reconstituted in the ring-shaped
chamber of the apparatus by compaction. The specimen was then consolidated under
a specific effective normal stress in a range of 100 kN/m$^2$ to 400 kN/m$^2$ until
consolidation was achieved. In this study, consolidation was completed when the
consolidation deformation was smaller than 0.01 mm within 24 hr (Kramer et al.,
1999; Shinohara and Golman, 2002). Then, the consolidated specimen is subjected to


shearing under constant normal stress by rotating the lower half of the shear box
attached to a gear, while the upper half remains still. In ring shear tests, the normal
stress at the shearing was the same as at consolidation stage. Shear strength of loess
specimen was recorded at intervals of 1s before the peak shear strength, after the peak,
the sampling rate was increased to 1 min.

In this study, ring shear tests were performed in a single stage under naturally

drained condition and the samples were subjected to shear until the residual state was
achieved. Drained condition of the shearing process is provided by two porous stones
attached on the top and the bottom platen of the specimen container. As for soil
specimens with low permeability, the rate of excess pore pressure generation in the
shear box may exceeded that of pore-pressure dissipation, this type of condition is
identified as naturally drained condition in previous studies(Okada et al., 2004).
Furthermore, Tiwari (2000) asserted that it was acceptable to use a shear rate below
1.1 mm/min to simulate the field naturally drained condition. Thus, shear rates of 0.1
mm/min and 1 mm/min were used in this study to simulate the naturally drained
condition of the slip zone soils.
**4.  Results**

Twenty -four specimens were tested to investigate the residual shear

characteristics of the saturated loess in the ring shear apparatus. Residual shear
strength of loess was determined following the research conducted by Bromhead
(1992) who pointed out that the residual stage is attained if a constant shear stress is
measured for more than half an hour. Tests results are shown in this section.



### 4.1. Shear behavior


Figs. 3(a)- 5(a) show the typical shear characteristics of the loess (shear rate of 0.1
mm/min and 1 mm/min) obtained from three different locations, where, the shear
stress is plotted against the shear displacement. It is a widely accepted fact that
normal stress has effect on the shear behavior of the soil (Stark Timothy et al., 2005;
Eid, 2014; Kimura et al., 2015; Wang et al., 2019), thus, the shear behavior of
samples at the peak and residual stages, where, the determined peak friction
coefficient as well as residual friction coefficient are plotted in Figs. 3(b)-5(b) against
the corresponding effective normal stresses as well. The friction coefficient is defined
as the shear stress divided by the effective normal stress.
Figs. 3(a)-5(a) demonstrate that shear stress increases dramatically within small
shear displacement and then reduces with shear displacement, until residual
conditions were achieved at large displacements. Furthermore, it is obvious that the
peak strength and the residual strength of samples with high shear rate are almost
smaller than that of the samples with low shear rate. It can be found that shear
displacement to achieve the residual stage for specimens with high shear rate is
greater than that of the low rate. For example, the minimum shear displacements for
attaining residual condition for Djg specimens with low and high shear rate were
about 360 mm and 650 mm, respectively. Under the shear rate of 0.1 mm/min and 1
mm/min, Ydg specimens need approximately 80 mm and 1,400 mm displacement to
achieve residual stage. However, Dbz specimens require about 40 mm and 60 mm
displacement to reach residual condition for low and high shear rate, respectively.

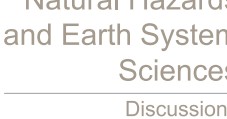 

In Figs. 3(a)- 5(a), a clear drop can be seen, at any normal stress, for specimens
obtained from all sites. It is obvious that Djg specimens showed greater peak-post
drop than that of Ydg and Dbz specimens. For example, at the normal stress of 100
$kN/m^2$, Djg samples show approximately 47.3% and 36.8% decrease from the peak
friction coefficient to the residual friction coefficient at low and high shear rates (Fig.
3(b)), respectively, which is greater than in the Ydg samples (about 9.8% and 10.3%
in Fig. 4(b)) and Dbz samples (about 2.4% and 3.2% in Fig. 5(b)). In Djg samples, an
obvious slickenside was observed on the shear surface (Fig. 6). This phenomenon
indicates a high degree of reorientation of platy clay minerals parallel to the direction
of shearing. In Figs. 3(b)- 5(b), on average, it was found that the decrease in the
friction coefficient from the peak strength in the Djg sample is almost 18.1% and
21.3% for the sample consolidated at normal stress of 400 $kN/m^2$ under the low and
high shear rate (Fig. 3(b)), while such reduction in friction coefficient in Ydg sample
are only about 4.1% and 4.8% (Fig. 4(b)). Furthermore, under the low and high shear
rate, the friction coefficient reduction in Dbz samples are only approximately 5.6%
and 6.0% (Fig. 5(b)). Skempton (1985) reported that the strength of soils falls to the
residual value in ring shear tests, owing to reorientation of platy clay minerals parallel
to the direction of shearing. Based on the conclusion that the post-peak drop in
strength of normally consolidated soil is only due to particle reorientation after the
peak strength (Skepmton, 1964; Mesri and Shahien, 2003), the results demonstrated
that the Djg landslide soil existed the greater particle reorientation compared with that
of other two landslide soils.




**4.2. Effect of normal stress on the friction coefficients**

It can be seen from the Figs. 3(b)-5(b) that the friction coefficients (peak and residual) are higher at low effective normal stress levels compared with that at high normal stress. For example, with normal stress increasing from 100 kN/m$^2$ to 400 kN/m$^2$, the peak and residual friction coefficient of Djg landslide soils at the shear rate of 0.1 mm/min reduce from 0.569 to 0.32 and from 0.3 to 0.262 (Fig. 3(b)), respectively. Similarly, results obtained from other two landslides loess also show that the friction coefficients decrease nonlinearly with normal stresses (Figs. 4(b) and 5(b)). Furthermore, specimens with shear rate of 0.1 mm/min attained greater friction coefficients than that with shear rate of 1 mm/min (Figs. 3(b)-5(b)).

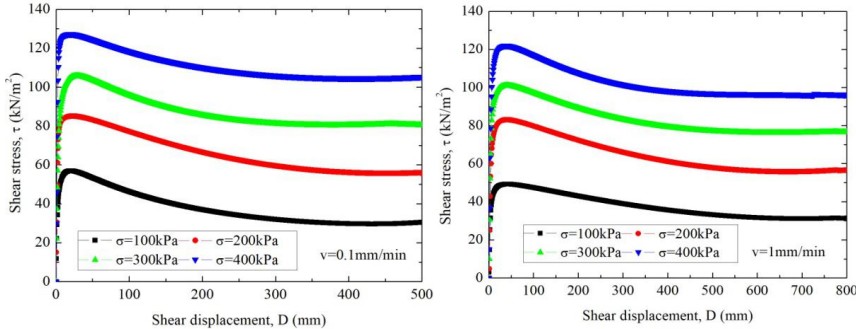


(a)Relationship between shear stress and shear displacement

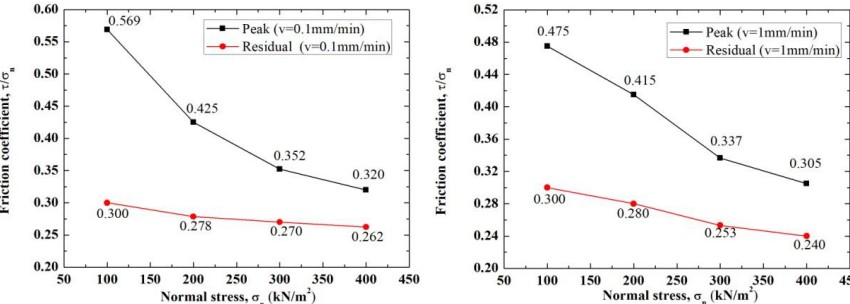


(b)Relationship between friction coefficient and normal stress



Figure 3. Shear behavior characteristics of Djg soil samples

(a)Relationship between shear stress and shear displacement

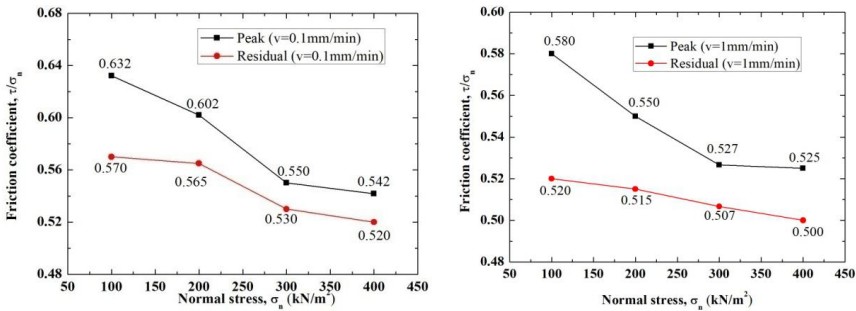

(b)Relationship between friction coefficient and normal stress
Figure 4. Shear behavior characteristics of Ydg soil samples

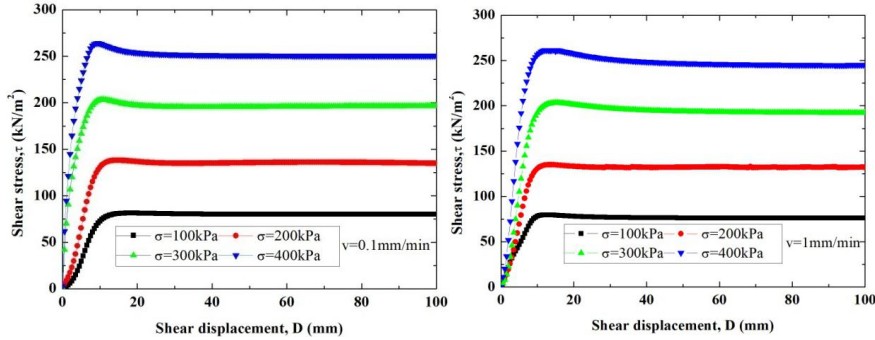

(a) Relationship between shear stress and shear displacement





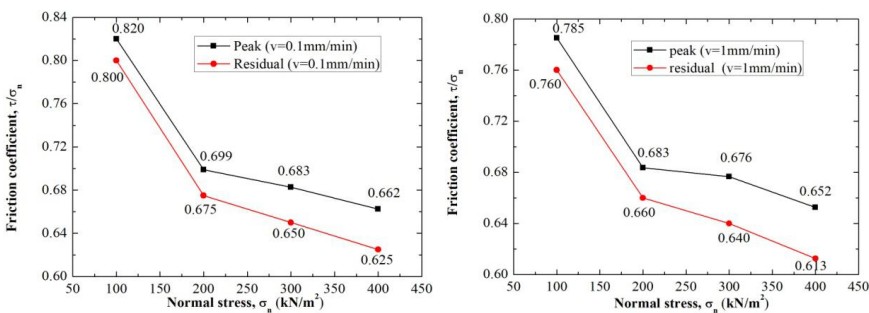

(b) Relationship between friction coefficient and normal stress

Figure 5. Shear behavior characteristics of the Dbz soil samples

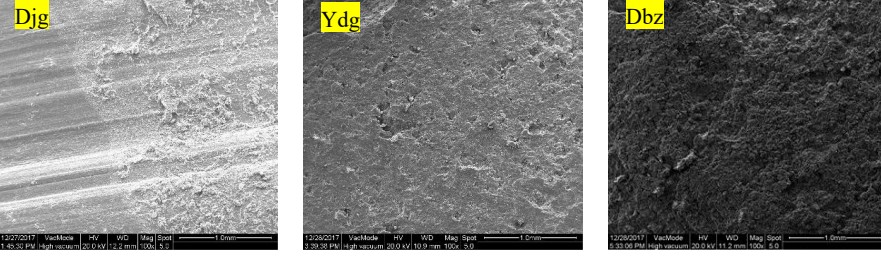

Figure 6. SEM photographs of the shear surface of loess samples (100 magnification)

**4.3. Effects of shear rate on residual strength parameter**

For the samples described above, Figs. 7-9 show the relationships between the residual friction coefficient and the normal stress, and the residual strength parameters. The residual friction coefficient is plotted against the normal stress. The residual friction coefficient is defined as the residual shear strength divided by normal stress. It has been recognized that the shear strength parameters including cohesion and friction angle (Terzaghi, 1951; Stark Timothy et al., 2005). However, according to the previous studies, the residual angle of soils varies depended on the soil properties as well as the magnitude of normal stress provided the residual cohesion of soil is zero (Skempton, 1964; Bishop; Kimura et al., 2014). Thus, in this study, the residual frictions are calculated by Coulomb's law assumed the residual cohesion is zero



following the previous studies (Skempton, 1985). The residual strength parameters
were defined as $\phi_r$ (0.1) and $\phi_r$ (1) at the low shear rate and high shear rate,
respectively. And the difference between the residual friction angles at two shear rates
was defined as $\phi_r$ (1) - $\phi_r$ (0.1). Comparatively, the residual friction coefficient was
defined as $\tau_r/\sigma_n$ (0.1) at the low shear rate and $\tau_r/\sigma_n$ (1) at the high shear rate,
respectively. Furthermore, the difference between the residual friction coefficients
was defined as $\tau_r/\sigma_n$ (1) - $\tau_r/\sigma_n$ (0.1). Table 2 summarized the residual shear
parameters of the landslide soils.
Fig. 7 shows that the residual friction coefficients are relatively low in Djg
samples. The coefficients $\tau_r/\sigma_n$ (0.1) and $\tau_r/\sigma_n$ (1) at the normal stress of 100 kN/m$^2$
to 400 kN/m$^2$ ranged from 0.3 to 0.262 and from 0.3 to 0.24, respectively. The
difference between the friction coefficients, $\tau_r/\sigma_n$ (1)- $\tau_r/\sigma_n$ (0.1), at each normal
stress level are varied in a range of -0.022 to +0.002. For the difference between the
residual friction angles, $\phi_r$(1)- $\phi_r$(0.1), ranged from -1.212° to +0.079° (Table 2). For
normal stress above 200 kN/m$^2$, the residual friction coefficient $\tau_r/\sigma_n$ (0.1) was found
to be greater than the residual friction coefficient $\tau_r/\sigma_n$ (1). For this sample, residual
friction coefficients show a slight decrease with the shear rate for normal stress above
200 kN/m$^2$.



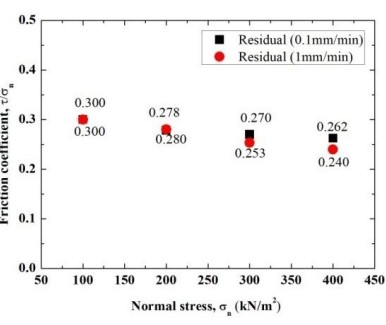 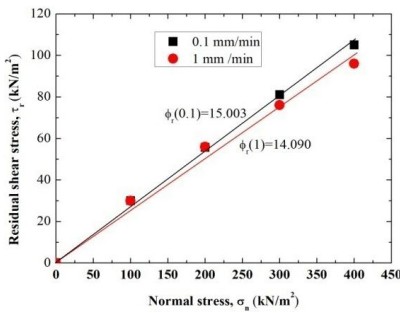

Figure 7. Relationships between residual shear stress and normal stress, and residual strength parameter for Djg soil sample

Fig. 8 gives the results of the Ydg samples. The coefficients $\tau_r/\sigma_n(0.1)$ and $\tau_r/\sigma_n$ (1) under the normal stress of 100 kN/m$^2$ to 400 kN/m$^2$ ranged from 0.57 to 0.52 and from 0.52 to 0.50, respectively. Furthermore, the difference $\tau_r/\sigma_n(1)$- $\tau_r/\sigma_n(0.1)$ at each normal stress was from -0.05 to -0.02. As for the difference between the residual friction angles, $\phi_r(1) - \phi_r(0.1)$, was in a range of -2.218° to -0.909°. In case of Ydg soil sample, the residual friction coefficients decreased with increase of shear rate for all normal stress levels.

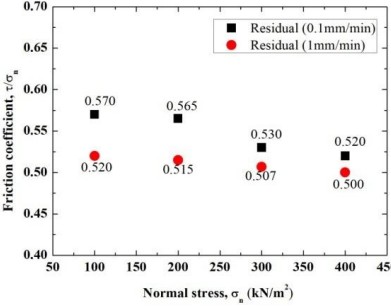 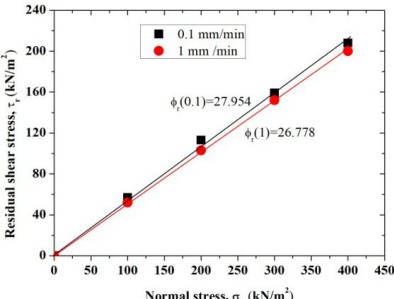

Figure 8. Relationships between residual shear stress and normal stress, and residual strength parameter for Ydg soil samples

Fig. 9 presents the results of the Dbz samples. The coefficients $\tau_r/\sigma_n(0.1)$ and $\tau_r/\sigma_n$ (1) at the normal stress of 100 kN/m$^2$ to 400 kN/m$^2$ ranged from 0.8 to 0.625 and



from 0.76 to 0.613, respectively. The difference $\tau_r/\sigma_n$ (1)- $\tau_r/\sigma_n$ (0.1) at each normal
stress was from -0.04 to -0.01. The difference $\phi_r$(1)- $\phi_r$(0.1) was from -1.425° to
-0.405°. For Dbz samples, there was somewhat decrease tendency of the residual
friction coefficients with the increasing of the shear rate for all normal stress levels. It
is noted that the maximum difference was found at the lowest normal stress of 100
kN/m$^2$.

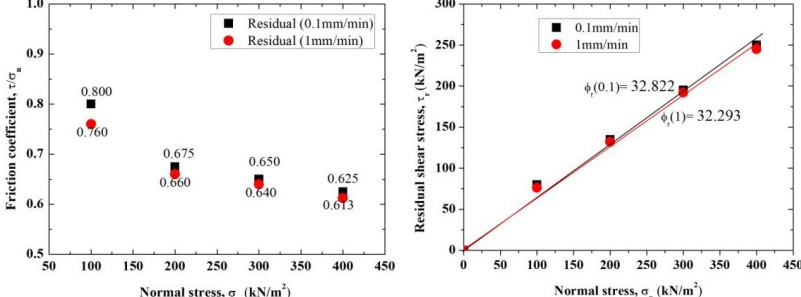

Figure 9. Relationships between residual shear stress and normal stress, and residual
strength parameter for Dbz soil sample

Table 2 summarizes residual strength parameters including $\phi_r$ (0.1) and $\phi_r$ (1) of

all specimens obtained from the ring shear tests in this study. As for the Djg samples,
the residual strength parameter $\phi_r$(0.1) and $\phi_r$(1) for all normal stress were found to
be 15.003° and 14.09° (Fig. 7), respectively. However, the residual friction angles $\phi_r$
(0.1) and $\phi_r$ (1) of the Ydg samples were obtained to be 27.954 ° and 26.778° (Fig. 8),
respectively. In the case of Dbz sample, the friction angles $\phi_r$ (0.1) and $\phi_r$ (1) were
high, 32.822° and 32.293° (Fig. 9), respectively. The residual friction angles $\phi_r$ (0.1)
and $\phi_r$ (1) under all normal stresses were from 15.003° to 32.822° and from 14.09° to
32.293°, respectively.

Due to the influence of the shear rate, the difference $\phi_r$ (1) - $\phi_r$ (0.1) in the Djg,





Ydg and Dbz samples, were -0.913°, -1.176° and -0.529° , respectively. Wang (2014)
and Fan et al. (2017) asserted that the residual shear strength of remolded loess hardly
affected by shear rate below 5 mm/min. However, the results in this study shown that
$\phi_r(1)$ - $\phi_r(0.1)$ under all normal stress levels were negative for loess. Moreover, the
absolute value of $\phi_r(1)$- $\phi_r(0.1)$ in Ydg samples even reached up to 1.176°.

Table 2 Residual shear strength parameter of landslide soils

| No | Sample | Normal stress(kN/m²) | Residual strength parameter | | | | Difference in parameter | |
|---|---|---|---|---|---|---|---|---|
| | | | 0.1  mm/min  $\phi_r$ (0.1)  ($c_{r(0.1)}$=0) (Degrees) | | 1   mm/min  $\phi_{r(1)}$  ($c_{r(1)}$=0) (Degrees) | | $\phi_{r(1)}$- $\phi_{r(0.1)}$  (Degrees) | |
| | | | Under each $\sigma_n$ | Under all $\sigma_n$ | Under   each $\sigma_n$ | Under all $\sigma_n$ | Under each $\sigma_n$ | Under all $\sigma_n$ |
| 1 | Djg | 100 | 16.699 | 15.003 | 16.699 | 14.090 | 0 | -0.913 |
| | | 200 | 15.563 | | 15.642 | | 0.079 | |
| | | 300 | 15.110 | | 14.216 | | -0.894 | |
| | | 400 | 14.708 | | 13.496 | | -1.212 | |
| 2 | Ydg | 100 | 29.683 | 27.954 | 27.474 | 26.778 | -2.209 | -1.176 |
| | | 200 | 29.466 | | 27.248 | | -2.218 | |
| | | 300 | 27.923 | | 26.870 | | -1.053 | |
| | | 400 | 27.474 | | 26.565 | | -0.909 | |
| 3 | Dbz | 100 | 38.660 | 32.822 | 37.235 | 32.293 | -1.425 | -0.529 |
| | | 200 | 34.019 | | 33.425 | | -0.594 | |
| | | 300 | 33.024 | | 32.619 | | -0.405 | |
| | | 400 | 32.005 | | 31.487 | | -0.518 | |


**4.4. Influence of the shear rate on the residual friction angles according to soil**
**properties**


It has been recognized that residual shear strength of soils is closely related with

soil properties, such as particle size distribution (PSD), liquid limit (LL), plasticity

index (Ip)and clay fraction (CF) (Terzaghi et al., 1996). Fig. 10 depicts the

relationships between residual friction angles as well as the difference in the residual

friction angles and soil properties including liquid limit (LL), plasticity index (Ip) and

clay fraction (CF) at two shear rates. The residual friction angles at two shear rates

decreased nonlinearly with the increasing of the LL. As for the relationship between

the $\phi_r$ and Ip, the $\phi_r$ under the low and high shear rates decreases from about 32° to 15°

with increasing the Ip from 11 to 16. These findings agree well with the early studies

(Wesley, 2003; Tiwari et al., 2005). With increasing of CF from 9% to 24%, the

residual friction angles under low and high shear rates were found to decrease (Fig.

10). These observations are consistent with previous studies (Lupini et al., 1981; Gibo

et al., 1987). Interestingly, for Dbz and Ydg soils of which have similar percentage of

clay fraction, the residual friction angles at both shear rates varied. However, in the

relationships between the difference in the residual friction angles and the soil

properties, no clear correlations were found.

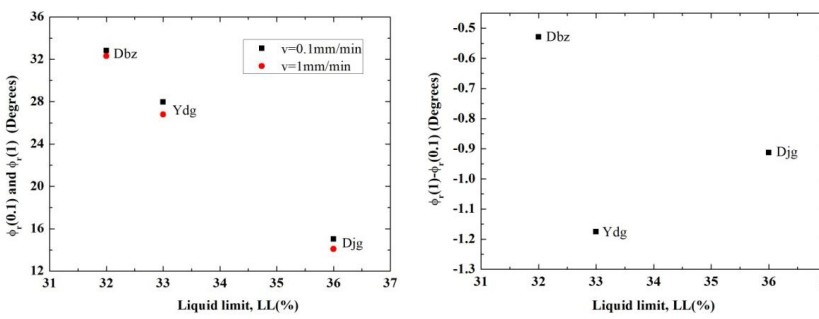



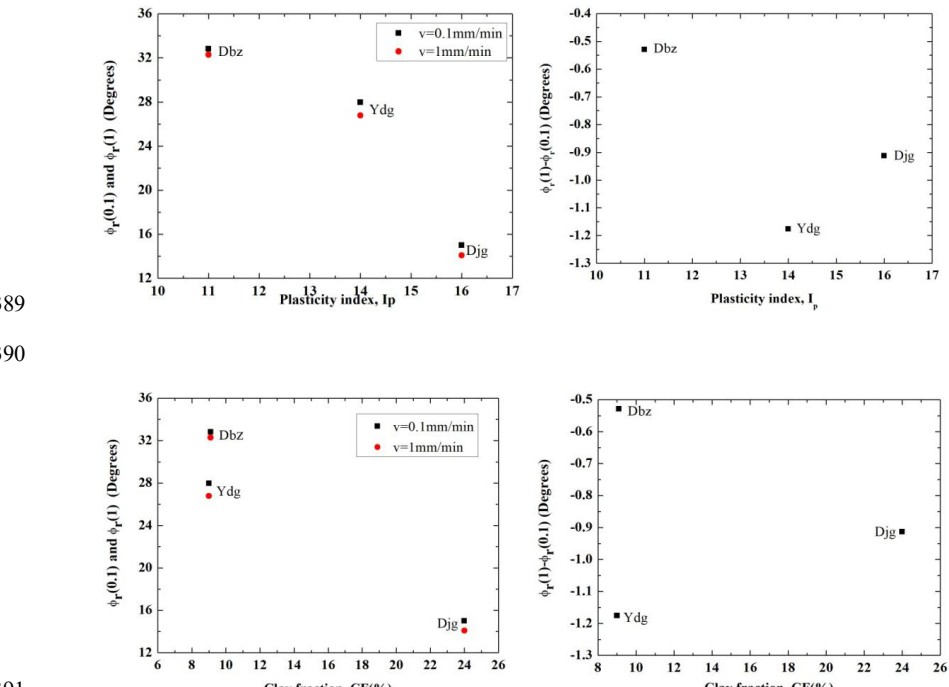




Figure 10. Relationships between residual shear parameter, the difference in
residual shear parameter and the soil properties at two shear rates
**5.**  Discussion
Examination of the ring shear test results provides a basis for some general
comments on the use of tests results with different shear rates, partially deepening
some aspects deriving from previous studies.
From the experimental results on the three selected landslides, it was found that
there is a negative relationship between residual friction coefficients and shear rates
for all samples (Figs. 7, 8 and 9). Such a negative effect of shear rate (higher residual
friction coefficients at lower rates) has been reported in the literature for fine-grained
soils (Tika et al., 1996; Gratchev Ivan and Sassa, 2015). This effect may be closely
associated with ability of clay particles in specimen to restore broken bonds at




different shear rates. Previous studies (Osipov et al., 1984; Perret et al., 1996).
concluded that with higher shear rates, the breakdown of the bonds between clay
particles or flocs exceeds the restoration bond, leading to reduction in residual friction
coefficients. In contrast, the bonds between particles are rebuilt quickly and the
recovery rate can catch up the breakdown rate at lower shear rates. Therefore, the
weaker bonding between particles could explain the strength drop with the increasing
of the shear rate in this study.

The difference between the friction coefficients, $\tau_r/\sigma_n(1)$- $\tau_r/\sigma_n(0.1)$, at each

normal stress level varies in different locations. $\tau_r/\sigma_n(1)$- $\tau_r/\sigma_n(0.1)$ in Ydg specimen
are greater compared with that in Djg and in Dbz specimen (Table 2). As for Ydg and
Dbz specimen, it is found that the shear rate effect on the friction coefficient can be
seen to decrease with normal stress (Figs. 8 and 9). By contrast, there is an increasing
tendency in the influence of shear rate on the friction coefficient with normal stress in
Djg specimen (Fig. 7). Gibo et al. (1987) reported that the residual friction angle of
soils was controlled by the effective normal stress as well as by the CF. Interestingly,
Ydg (with CF 9%) and Dbz (with CF 9.1%) specimens with almost the same fraction
of clay showed similar shear rate effect on the residual friction coefficient with
normal stress increasing, however, Djg (with 24% CF) showed the contrast tendency
of shear rate effect on residual friction coefficient with normal stress, indicating that
such effect is closely associated with CF. Therefore, as for Ydg and Dbz with
relatively low fraction of CF, there is an increase effect of shear rate on residual
friction coefficient with decreasing of normal stress. Thus, for the application of





measured residual friction coefficient for stability analysis of shallow landslides with
lower overburden pressure, it is significant for us to use a low shear rate in ring shear
tests to measure residual shear strength parameters. On other hand, for Djg with high
CF, it is more reliable to use a low shear rate in ring shear tests to determine residual
friction coefficient for stability analysis of deep landslides with high overburden
pressure.

**6.  Conclusion**

A series of ring shear tests were conducted on loess obtained from three landslides

to study the residual shear characteristics of saturated loess. Based on the test results,
the effect of the shear rate on the residual shear characteristics of loess in naturally
drained condition was examined. The following conclusions can be drawn:
1.   Ring shear test revealed that (i) shear displacement to achieve the residual stage

with high shear rate is greater than that of the low shear rate; (ii) Both the peak

and residual friction coefficient became smaller with increase of shear rate for

each sample;(iii) The greater difference between the peak and the residual friction

coefficient in loess samples could be attributed to relatively well-developed

slickenside on the shear surface.

2.   At the two shear rates, there was a nonlinearly decrease trend of the residual

friction coefficient with the normal stress in all loess samples. The difference

between the friction coefficients, $\tau_r/\sigma_n(1)$- $\tau_r/\sigma_n(0.1)$ was found to decrease

with normal stress in Ydg and Dbz specimens while increase with normal stress in

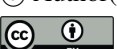


Djg specimens, indicating that CF may be closely associated with shear rate effect
on residual friction coefficient with normal stress.
3.  The difference at the two shear rates, $\phi_r(1)$ - $\phi_r(0.1)$, under each normal stress
level were either negative or positive. However, under all normal stress, the
difference at the two shear rates $\phi_r(1)$ - $\phi_r(0.1)$ was found to be negative. Such
negative shear rate effect on loess could be attributed to greater ability of clay
particles in specimen to restore broken bonds at low shear rates.
4.  The relationships between the $\phi_r$ under two shear rates and soil properties (LL, Ip),
demonstrated that the $\phi_r$ at both shear rates decreased gradually with the
increasing of LL and Ip. However, no clear correlations between the difference in
the $\phi_r$ at low and high shear rates and the soil properties were found.




**Code availability:** Code can be made available by the authors upon request.
**Data availability:** Data can be made available by the authors upon request.
**Author contributions:** BL,JP and QH conceived and designed the method; BL
produced the results, and wrote the original manuscript under the supervision of JP.
JP and QH writing-review and editing.
**Competing interest:** The authors declare that they have no conflicts of interest.
**Acknowledgments:** This research was supported by the Major Program of National
Natural Science Foundation of China (Grant No. 41790440), the National Natural
Science Foundation of China (No.41902268) and the China Postdoctoral Science
Foundation (No. 2019T120871).





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
