# Peer review of "1. Introduction"

_Natural Hazards and Earth System Sciences, 2019_

## Referee Comment (RC1) · Anonymous Referee #1 · 8 Oct 2019

This paper provides interesting data on soils obtained from three landslides and could be of interest to many readers. However, I feel some additional work is required prior to it being suitable for publication in the journal.

1.Some more details need to be provided on the soils (Djg, Ydg and Dbz), e.g. particle size distribution curves. 2.Line 91, "their relationships" could be changed to "the relationship between the residual strength parameters". 3.Line 146, "crushed", does this affect results? 4.Line 169, to keep consistency with the text body, change "moisture water content" to "moisture content", please revise it. 5.Lines 172-173, there are 2 main types discussed in the literature, the Bromhead device and the IC/NGI device, which one is this? Please point out it in the paper. 6.Line 198, please give more detail about compaction. 7.It seems that you do not need to mention the

shearing process in lines 203-204 again since you have mentioned the procedure in Lines 176-177. 8.Line 207, "the sampling rate was increased to 1 min", please check the sampling rate unit. 9.Line 209, in my opinion, "the samples were subjected to shear" could be changed to "the samples were subjected to shearing". 10.Lines 209-210, how do you define the residual state was achieved? 11.Lines 238-239, The authors should clearly define what are low and high shear rate. 12.Lines 375-376, the authors do not need to write Liquid limit (LL) again since you have mentioned that in lines 372-373, just use LL in lines 375-376. 13.Line 400, change "Figs. 7, 8 and 9" to "Figs. 7-9", please revise it. 14.In Table 1, units missing on the header. Feel PSD curve is necessary. Please revise it. 15.The use of the English language needs some work. I really recommend the authors to send the manuscript to be reviewed thoroughly by a native English speaker.

Please also note the supplement to this comment:
https://www.nat-hazards-earth-syst-sci-discuss.net/nhess-2019-156/nhess-2019-156-RC1-supplement.pdf

---

## Short Comment (SC1) · 19 Oct 2019

Residual shear strength of soils is an important soil parameter for assessing the stability of landslides. However, compared with the results of tests on clay or sand, understanding of the shear characteristics of silty soil, such as loess, is not yet complete. To clarify the residual shear characteristics of loess under the effect of the shear rate, a series of naturally drained ring shear tests were conducted on loess obtained from three landslides at two shear rates and the residual shear characteristics of loess at the residual state was examined. The research would be helpful for evaluating the stability for the slip surface of first-time landslides as well as reactivated landslides. About language writing, the corrections of spell seem not necessary. However, a moderate revision is recommended for this paper.

[Figure]

General comments:

ïĆůïÅă In the "abstract" section and "conclusion" section, the authors point out "such negative shear rate effect on loess could be attributed to a greater ability of clay particles in specimen to restore broken bonds at low shear rates." In my opinion, since you do not have data or results to support such conclusion, I suggest you to delete the sentence or revise it as follows: "Such negative shear rate effect on loess may be attributed to a greater ability of clay particles in specimen to restore broken bonds at low shear rates".

ïĆůïÅăIn "Geological setting of landslide sites" section, you should use a more concise description for three landslides. In my opinion, the authors do not need to focus much on the background of landslides such as the volume, width of landslides. However, you should put more emphasis on the soil properties.

ïĆůïĆůI only found the physical parameters of slip-zone loess in Table 1, please add more information about the soil, such as the particle size distribution curves of soil.

ïĆůïÅă Concerning the reference list: In "testing sample" section, the soil samples were crushed with a mortar and pestle. Please kindly cite 1-2 references to support it.

ïĆůLines 205-207, please explain the reason why you use different intervals (sampling rates) to record the shear strength of soil. Does this affect the experimental results?

ïĆůLines 238-239: what is "high shear rate" and "low shear rate" meaning? You have to point out their meaning when it appears in the manuscript for the first time.

ïĆůïÅăPlease add "the limitation of this study" in the "Discussion" section since slip surface soils used in this study are obtained from only three landslide sites.

Minor comments:

Lines 114-115: delete "with the height difference between the toe and the top of landslide about 55 m" since the authors have introduced the top and the toe altitude of the

landslide.

Line 120: change "ranged" to "ranging".

Line 125: change "quaternary" to "Quaternary".

Line 139: change "the Figure 1" to "Figure 1".

Line 185: change "be monitored by computer in shearing process" to "be monitored by a computer in the shearing process".

Line 192: It seems that "the" is missing before the word "present".

Line 226: change "shear rate" to "the shear rate".

Lines 236-237: "residual conditions were achieved at large displacements" should be "the residual condition was achieved at large displacements". Please revise it.

Line 335: change "ÑĎr (0.1)" to "ÑĎr (0.1)".

Lines 455,456,458: change "ÑĎr" to "ÑĎr". Please check this issue thoroughly in the manuscript.

Please also note the supplement to this comment:
https://www.nat-hazards-earth-syst-sci-discuss.net/nhess-2019-156/nhess-2019-156-SC1-supplement.pdf

---

## Referee Comment (RC2) · Anonymous Referee #2 · 16 Dec 2019

This paper deals with the effect of the shear rate on the residual shear strength of loess from three landslides by using a ring shear apparatus. Overall, this is an interesting manuscript because the topic can be considered of large significance for international researches in the field; however, this manuscript needs some important improvements to get it into a position to be acceptable for publication. Thus, a major revision is recommended. My critical review is summarized in the following sentences: - The title could be more informative although it is pertinent and understandable; - The abstract should be more precise and clear, although the most important results have been mentioned (Please, find the file attached for details). Authors should better emphasize the aim, importance and results of this study, and why it should be considered as relevant to be published in an international journal; - The introduction

provides relevant background information. Important scientific publications, on which the work is based, are cited but some recent original papers are not considered; - Geological setting and sampling sites if, on the one hand, require a brief description, on the other hand, should contain all the useful information for the purpose of the work. Congruent bibliographic references are missing. Please, find the file attached for details; - Description of the materials used (grain size distribution, percentage and mineralogy of the clay fraction, plasticity of fine) is very important and can explain some discrepancies between different interpretations. Please, find the file attached for details; - Description of the method used in this study should be detailed and complete. What does it mean for low or high shear rate and low or high effective normal stress? Please, find the file attached for details; - Results and discussion may be combined into a single section to avoid repetitions in the discussion, which would thus be more interesting and complete, also with references to earlier or contemporary studies relevant to the topic. Discussion of the results should include aspects related to dilatancy and critical state; - Conclusions summarize the main findings of the experimental research but could describe their significance or implication, in light of what was already known about the subject of the study, and present fresh insights or possible new ways of approaching research questions; - Text, tables, citations and references should be formatted according to the journal's instructions; - A thorough revision of the text with the help of a native English speaker is suggested.

Please also note the supplement to this comment:
https://www.nat-hazards-earth-syst-sci-discuss.net/nhess-2019-156/nhess-2019-156-RC2-supplement.pdf

**Supplement:**

[revised manuscript text omitted]

---

## Author Comment (AC1) · 4 Feb 2020

This paper provides interesting data on soils obtained from three landslides and could be of interest to many readers. However, I feel some additional work is required prior to it being suitable for publication in the journal. Reply: Thank you for your encouraging comments on our work.

1. Some more details need to be provided on the soils (Djg, Ydg and Dbz), e.g. particle size distribution curves. Reply: Implemented. See Figure 2 of the revised manuscript.

2. Line 91, "their relationships" could be changed to "the relationship between the residual strength parameters". Reply: Implemented. See lines 98-99 of the revised manuscript.

3. Line 146, "crushed", does this affect results? Reply: Implemented. The procedure "crushed" would not affect results. The purpose of crushing with a mortar and pestle is to to disintegrate aggregate. Crushing samples has been found suitable to determine the residual strength of the remoulded soils (Stark et al., 2005). This should be done with care so as not to destroy silty-dominated loess. See details in lines 154-159 of the revised manuscript.

4. Line 169, to keep consistency with the text body, change "moisture water content" to "moisture content", please revise it. Reply: Implemented. See line 184 of the revised manuscript.

5. Lines 172-173, there are 2 main types discussed in the literature, the Bromhead device and the IC/NGI device, which one is this? Please point out it in the paper. Reply: Implemented. SRS-150 used in this study is a type of Bromhead ring shear apparatus. See lines 197-198 in the revised manuscript.

6. Line 198, please give more detail about compaction. Reply: Implemented. See lines 229-232 of the revised manuscript.

7. It seems that you do not need to mention the shearing process in lines 203-204 again since you have mentioned the procedure in Lines 176-177. Reply: Implemented. We have deleted the contents in original lines 203-204 according to the review's suggestion.

8. Line 207, "the sampling rate was increased to 1 min", please check the sampling rate unit. Reply: Implemented.

9. Line 209, in my opinion, "the samples were subjected to shear" could be changed to "the samples were subjected to shearing". Reply: Implemented. See line 241 of the revised manuscript.

10. Lines 209-210, how do you define the residual state was achieved? Reply: Implemented. In this study, following the Bromhead (1992), the residual state was defined

when a constant shear stress is obtained for more than half an hour. See lines 242-243 in the revised manuscript.

11. Lines 238-239, The authors should clearly define what are low and high shear rate. Reply: Implemented. See lines 272-274 of the revised manuscript.

12. Lines 375-376, the authors do not need to write Liquid limit (LL) again since you have mentioned that in lines 372-373, just use LL in lines 375-376. Reply: Implemented. See line 453 of the revised manuscript.

13. Line 400, change "Figs. 7, 8 and 9" to "Figs. 7-9", please revise it. Reply: Implemented. See line 403 of the revised manuscript.

14. In Table 1, units missing on the header. Feel PSD curve is necessary. Please revise it. Reply: Implemented. See Table 1 in the revised manuscript. With regards to PSD curve, we have added the PSD curves in the revised manuscript, see Figure 2 in the revised manuscript.

15. The use of the English language needs some work. I really recommend the authors to send the manuscript to be reviewed thoroughly by a native English speaker. Reply: Implemented. The revised manuscript has been reviewed thoroughly by a native English speaker to improve the grammar and readability.

---

## Author Comment (AC2) · 4 Feb 2020

-Reviewer #2

This paper deals with the effect of the shear rate on the residual shear strength of loess from three landslides by using a ring shear apparatus. Overall, this is an interesting manuscript because the topic can be considered of large significance for international researches in the field; however, this manuscript needs some important improvements to get it into a position to be acceptable for publication. Thus, a major revision is recommended. My critical review is summarized in the following sentences:

Reply: Thank you for your encouraging comments on our work.

1. The title could be more informative although it is pertinent and understandable

[Figure]

Reply: Implemented. The title has been changed as "Shear rate effect on the residual strength characteristics of saturated loess in naturally drained ring shear tests " according to the review's suggestion. See title in the revised manuscript.

2. The abstract should be more precise and clear, although the most important results have been mentioned (Please, find the file attached for details).

Reply: Implemented. See lines 29-35 of the revised manuscript.

3. Authors should better emphasize the aim, importance and results of this study, and why it should be considered as relevant to be published in an international journal.

Reply: Implemented. See lines 70-89 of the revised manuscript.

4. The introduction provides relevant background information. Important scientific publications, on which the work is based, are cited but some recent original papers are not considered.

Reply: Implemented. We have cited some recent original papers, see lines 45-46ïijŇ73-75ïijŇ 85 of the revised manuscript.

5. Geological setting and sampling sites if, on the one hand, require a brief description, on the other hand, should contain all the useful information for the purpose of the work.

Reply: Implemented. We have cited relevant references to describe the geological setting and sampling sites, see lines 110-111, 126, 132-133 of the revised manuscript.

6. Congruent bibliographic references are missing. Please, find the file attached for details

Reply: Implemented. See references in the revised manuscript.

7. Description of the materials used (grain size distribution, percentage and mineralogy of the clay fraction, plasticity of fine) is very important and can explain some discrepancies between different interpretations. Please, find the file attached for details

Reply: Implemented. See lines 71-76 of the revised manuscript.

8. Description of the method used in this study should be detailed and complete.

Reply: Implemented. See lines 218-220, 223-225, 229-232, 241-243 of the revised manuscript.

9. What does it mean for low or high shear rate and low or high effective normal stress? Please, find the file attached for details

Reply: Implemented. See lines 272-274, 313-315 of the revised manuscript.

10. Results and discussion may be combined into a single section to avoid repetitions in the discussion, which would thus be more interesting and complete, also with references to earlier or contemporary studies relevant to the topic.

Reply: Implemented. See lines 253, 263-264, 307-308, 341-348, 354, 401-423, 440-442 of the revised manuscript.

11. Discussion of the results should include aspects related to dilatancy and critical state

Reply: With regards to the dilatancy effect of the samples, we have added the relevant content in the manuscript, see lines 283-289 of the revised manuscript.

With regards to the critical state of the loess, we did not measure normal displacement of loess samples, therefore the critical state of samples was not discussed in the study.

12. Conclusions summarize the main findings of the experimental research but could describe their significance or implication, in light of what was already known about the subject of the study, and present fresh insights or possible new ways of approaching research questions

Reply: Implemented. See lines 488-496, 506-511 of the revised manuscript.

13. Text, tables citations and references should be formatted according to the journal's

instructions

Reply: Implemented. See Tables and references in the revised manuscript.

14. A thorough revision of the text with the help of a native English speaker is suggested.

Reply: Implemented. The manuscript has been revised with the help of a native English speaker.

---

## Author Response (AR2)

**Responses to the Comments by Editors and Reviewers**

Dear Professor Parise:

Upon your recommendation, we have carefully revised Paper nhess-2019-156 entitled "Shear rate effect on the residual strength characteristics of saturated loess" after considering all the comments made by the reviewers.

We thank Professor Parise, Editor and anonymous reviewers for their constructive comments. The manuscript has been significantly improved by incorporating their suggestions. The following are our point-to-point responses to their comments.

**Responses to the Comments from Editor and reviewers**
**Comments from the editors and reviewers:**

1. As indicated by the reviewer, there are some technical issues with formatting (tab.2, text justification, use of different font for references). Authors are kindly asked to adjust all these incongruences.

Reply: Implemented.

With regards to technical issues with formatting (Table 2), we have adjusted Table 2 according to the review's suggestion, see details in Table 2 (lines 455-456) of the revised manuscript.

With regards to technical issues with formatting (text justification), we have adjusted text, see details on lines 44-49, 90-100, 130-144, 154-167, 197-214, 219-259, 381-387, 402-424, 434-443, 459-474 and 484-522 of the revised manuscript.

With regards to technical issues with formatting (use different font for references), we have used the same font for references, see details on lines 568-570 of the revised manuscript.

2. Please also have a careful look at the text, where I have noticed there are many points where you need to separate words.

Reply: Implemented. See details on lines 33, 255, 261,283, 371, 373, 383, 394, 427, 440, 461, 496 of the revised manuscript.

[revised manuscript text omitted]

33, 72-74, 2006.